# Myoferlin Is a Yet Unknown Interactor of the Mitochondrial Dynamics’ Machinery in Pancreas Cancer Cells

**DOI:** 10.3390/cancers12061643

**Published:** 2020-06-21

**Authors:** Sandy Anania, Raphaël Peiffer, Gilles Rademaker, Alexandre Hego, Marc Thiry, Louise Deldicque, Marc Francaux, Naïma Maloujahmoum, Ferman Agirman, Akeila Bellahcène, Vincent Castronovo, Olivier Peulen

**Affiliations:** 1Metastasis Research Laboratory (MRL), GIGA-Cancer, Pathology Institute B23, University of Liège, B-4000 Liège, Belgium; sandy.anania@uliege.be (S.A.); r.peiffer@student.uliege.be (R.P.); g.rademaker@uliege.be (G.R.); naima.maloujahmoum@uliege.be (N.M.); f.agirman@uliege.be (F.A.); a.bellahcene@uliege.be (A.B.); vcastronovo@uliege.be (V.C.); 2Center for Interdisciplinary Research on Medicines (CIRM), Pathology Institute B23, University of Liège, B-4000 Liège, Belgium; 3Imaging Facilities, GIGA-Research, GIGA-Institute B36, University of Liège, B-4000 Liège, Belgium; alexandre.hego@uliege.be; 4Laboratory of Cellular and Tissular Biology, GIGA-Neurosciences, Cell Biology L3, University of Liège, B-4000 Liège, Belgium; mthiry@uliege.be; 5Institute of Neuroscience, Université catholique de Louvain, B-1348 Louvain-la-Neuve, Belgium; louise.deldicque@uclouvain.be (L.D.); marc.francaux@uclouvain.be (M.F.)

**Keywords:** myoferlin, mitochondria, mitofusin, pancreas cancer

## Abstract

Pancreas ductal adenocarcinoma is one of the deadliest cancers where surgery remains the main survival factor. Mitochondria were described to be involved in tumor aggressiveness in several cancer types including pancreas cancer. We have previously reported that myoferlin controls mitochondrial structure and function, and demonstrated that myoferlin depletion disturbs the mitochondrial dynamics culminating in a mitochondrial fission. In order to unravel the mechanism underlying this observation, we explored the myoferlin localization in pancreatic cancer cells and showed a colocalization with the mitochondrial dynamic machinery element: mitofusin. This colocalization was confirmed in several pancreas cancer cell lines and in normal cell lines as well. Moreover, in pancreas cancer cell lines, it appeared that myoferlin interacted with mitofusin. These discoveries open-up new research avenues aiming at modulating mitofusin function in pancreas cancer.

## 1. Introduction

Pancreas ductal adenocarcinoma (PDAC) is one of the deadliest diseases with a 5-year survival lower than 10%. In PDAC, mitochondria activity was described to be involved in tumor relapse [1] and in metastatic dissemination [2]. Mitochondria are highly dynamic structures oscillating from a branched network to individual organelles according to the metabolic needs or mitochondrial damages. This oscillation, referred as mitochondrial dynamics, was tightly linked to PDAC aggressiveness. Indeed, it was shown that increasing mitochondrial fusion suppressed oxidative phosphorylation (OXPHOS), promoted mitophagy, and improved median survival in PDAC mice models significantly [3]. Conversely, we previously showed in PDAC cell lines that a mitochondrial fission was associated with OXPHOS decrease, and enhanced autophagy [4]. Interestingly, in a mice model, a lower metastatic potential of PDAC cells was associated with a reduced OXPHOS [2]. Consequently, it appeared more important to disturb mitochondrial dynamics rather than to specifically inhibit fusion or fission [5]. Fusion or fission machineries are composed of a limited, but still growing, number of proteins among which the most studied are dynamin-related protein 1 (DRP1), optic atrophy 1 (OPA1), and mitofusins (MFN) 1 and 2 [6,7]. Post-translational modifications of these enzymes participate in the fine-tuning of mitochondrial dynamics. As an example, the initial step of fission is the recruitment of DRP1 at the mitochondrial surface, regulated by the phosphorylation ratio between serine 616 (S616) and S637. The S616 phosphorylation is a fission activator while the S637 one is a fission inhibitor [8].

Myoferlin is a ferlin family member protein, mainly known for its physiological function in membrane fusion, and its expression level was correlated with poor survival in several cancer types including PDAC [9]. Previously, our laboratory pointed to myoferlin as a potential biomarker in PDAC [10], where it is involved in essential membrane processes such as exocytosis and exosome production [11,12]. Using small interfering RNA (siRNA), we demonstrated that myoferlin depletion drives the cell toward mitochondrial fission, suggesting that this protein may act as a potential regulator of mitochondrial dynamics [4,13]. However, despite several advances, the mechanism of action of myoferlin remains largely unsolved. Interestingly, a myoferlin paralog, encoded by the dysferlin_v1 alternate transcript, was discovered to harbor a mitochondrial importation signal [14]. Moreover, a recent proteomic study undertaken in mice revealed that myoferlin could be present in mitochondria [15]. This discovery prompted us to further investigate the putative participation of myoferlin to the mitochondrial dynamics’ machinery. For the first time, we revealed that myoferlin interacts with mitofusins, and might contribute to their mitochondrial fusion activities [16].

## 2. Results

### 2.1. Endogenous Myoferlin Is Present in Mitochondrial Crude Extract and Colocalized Partly with Mitochondria

Previously, we showed that myoferlin silencing impaired mitochondrial network in Panc-1 cells [4]. Thus, we decided to use the same cell line to investigate the potential mitochondrial localization of endogenous myoferlin (Figure 1A). Using differential centrifugation steps, we prepared a mitochondrial crude extract. The abundance of a mitochondrial-specific 60 kDa protein (clone 113-1) [17] indicated a 4.7-fold enrichment factor in comparison to whole cell extract. Interestingly, the mitochondrial crude extract contained several myoferlin isoforms with a 1.6-fold increase compared to the whole cell extract. Even if its relative abundance decreased by 30%, GRP78 was still detectable in the mitochondrial crude extract indicating either a microsomal contamination or a GRP78 mitochondrial localization [18]. These encouraging results prompted us to perform an immunofluorescence staining to explore myoferlin localization inside Panc-1 cells. For this purpose, we used a goat polyclonal myoferlin antibody (K-16). As previously reported [19], myoferlin staining appeared as a punctuated signal spread all over the cytoplasm but with a higher density close to plasma membrane (Figure 1B). Correlative colocalization analysis of deconvoluted images revealed a partial colocalization between myoferlin and mitochondrial signals (Figure 1C). While linear correlation coefficients (PCC and SRCC) showed only a weak association (~0.20), Manders’ colocalization coefficients (M1 and M2) indicated an intermediate colocalization (>0.50). M1 describes the proportion of myoferlin pixels co-occurring with mitochondrial pixels, and vice-versa for M2. Several myoferlin-positive structures were identified in contact with mitochondria (Figure 1D,E). Interestingly, some myoferlin staining was located between mitochondrial sections close to each other and considered as potential mitochondrial fusion sites (Figure 1D,E). Myoferlin and mitochondrial colocalization was confirmed by “distance between objects”-based methods (Figure 1F). These methods showed >5% of the myoferlin-positive objects (*N* = 4286) were colocalized with a mitochondrial-object (*N* = 459) with a mean distance of 2 pixels, ranging from 0 to 5 pixels.

Immunofluorescence results were confirmed using an additional myoferlin polyclonal antibody raised in rabbits (Appendix A).

### 2.2. Endogenous Myoferlin Colocalized with Mitochondrial Fusion Machinery in Pancreas Cancer Cell Lines

Owing to the known function of myoferlin in membrane fusion, we thought to evaluate the colocalization of myoferlin with a component of the fusion machinery: mitofusins. We thus performed immunofluorescence using myoferlin antibody (K-16) and MFN1 antibody (H-65). In Panc-1 cells, myoferlin was mainly associated with MFN1 in the perinuclear region (Figure 2A). Linear correlation coefficients (Figure 2B) showed a strong association between stainings. “Distance between objects”-based methods (Figure 2C) revealed that 20% to 30% of the myoferlin-positive objects (*N* = 7128) were colocalized with a MFN1-positive object (*N* = 369) with a mean distance of 3 pixels, ranging from 0 to 5 pixels. These results were confirmed by using an additional myoferlin antibody raised in rabbit and a MFN1/2 polyclonal antibody (3C9) raised in mouse (Appendix A). In order to confirm these results, we performed a proximity ligation assay on Panc-1 cells. This experiment showed 21.3 ± 6.8 proximity dots per cell, indicating a maximal 40 nm distance between myoferlin and MFN1/2 (Figure 2D). We next inhibited myoferlin expression using siRNA to confirm the specificity of the proximity ligation assay signal. Myoferlin silencing suppressed more than 95% of the colocalization signal confirming the specificity of the colocalization (Figure 2E). Proximity ligation assay results were confirmed in Panc-1 cells by indirect fluorescence resonance energy transfer analysis showing a significant FRET ratio (Appendix A).

We then decided to confirm MFN1/2-myoferlin colocalization in three additional PDAC cell lines (BxPC-3, MiaPaCa-2 and PaTu8988T) for which we reported the relative myoferlin and MFN1/2 expression (Appendix A). In every evaluated cell line, a colocalization was identified mainly thanks to Manders’ colocalization coefficients (M1, M2) (Figure 3A–C). In these cell lines, linear correlation coefficients (PCC and SRCC) showed a weaker association (from 0.20 to 0.6 depending of the cell line) than in Panc-1 cell lines. In the BxPC-3 cell line, immunofluorescence staining showed that MFN1/2-myoferlin association was mainly localized at cell periphery (Figure 3A—left panel). In a PaTu8988T cell line, even if colocalization coefficients were sound, the very limited observable cytoplasm area and the low myoferlin expression level made results difficult to interpret. Consistent with the immunofluorescence results, proximity ligation assay showed less colocalization dots in BxPC-3, MiaPaCa-2, and PaTu8988T than in Panc-1 cells (Figure 3D). The relative amount of proximity dots appeared to be correlated with myoferlin abundance of each cell line. Considering our findings, we then tested whether myoferlin was physically interacting with MFN1/2.

### 2.3. Myoferlin Interacts with Mitofusins in Pancreas Cancer Cells

We first took advantage of an overexpression model of hemagglutinin (HA)-tagged myoferlin in Panc-1 cells to maximize the myoferlin-MFN1/2 interaction, and performed a coimmunoprecipitation assay. We immunoprecipitated MFN1/2 and showed the coprecipitation of HA-tagged myoferlin (Figure 4A). Encouraged by this unforeseen result, we decided to confirm the myoferlin-MFN1/2 interaction in endogenous expression systems. We thus performed the same experiment in Panc-1, BxPC-3, MiaPaCa-2 and PaTu8988T cell lines. In all cell lines tested, myoferlin coprecipitated with MFN1/2 (Figure 4B,4C) with an abundance in agreement with the previously described myoferlin expression level (Appendix A). In the light of our results, we wondered if the myoferlin-MFN1/2 interaction occurred in normal cells.

### 2.4. Myoferlin Colocalizes but Does Not Interact with Mitofusins in Normal Cells

Myoferlin expression is supposed to be low in differentiated normal cells. We thus selected subconfluent (90%) murine C2C12 myoblasts for their known functional expression of myoferlin [20] and immortalized human pancreatic normal epithelial (HPNE) cell lines with undifferentiated phenotype [21]. Immunofluorescence revealed a colocalization between myoferlin and MFN1/2 (Figure 5A–D). However, in these cell lines, MFN1/2 immunoprecipitation did not reveal a convincing physical interaction with myoferlin (Figure 5E,F) suggesting an interaction specific to cancer cells.

### 2.5. Mitochondrial Impact of Myoferlin Depletion in Pancreas Cancer Cells

In order to illustrate the functional role of myoferlin in mitochondria of pancreas cancer cells, we investigated mitochondrial network, mitochondrial ultrastructure and oxygen consumption rate (OCR) in myoferlin-depleted Panc-1 cells. Tetramethylrhodamine ethyl ester (TMRE) staining showed a mitochondrial swelling and a disruption of the mitochondrial network upon myoferlin silencing (Figure 6A). In Panc-1 cells transfected with irrelevant siRNA, ultrastructural analysis revealed elongated or circular mitochondrial sections with homogeneous matrix and well-defined cristae (Figure 6B). When Panc-1 cells were transfected with myoferlin siRNA, mitochondrial matrix appeared condensed with less abundant cristae. Oxygen consumption rate, reflecting electron transport chain activity, was significantly decreased by myoferlin silencing in Panc-1 cells (Figure 6C).

## 3. Discussion

Myoferlin is a member of the ferlin family, mainly known for its function in myoblast membrane fusion and membrane reparation. Thanks to its multiple C2 domains, myoferlin participates in the tethering of vesicles to membranes and to the calcium sensing. An extensive and excellent review has been recently published on the functions of ferlins in vertebrates [22]. Myoferlin expression is increased in a large panel of cancer cells and tumors, where most of the studies described its role in the recycling of membrane receptors (EGFR, IGFR, …). Its participation in signaling pathways explains why its depletion leads to a decrease of cell growth or migration [9,23].

Mitochondrial dynamics is closely related to human diseases such as Charcot–Marie–Tooth disease type 2A and optic atrophy. In cancer, an increasing body of literature links mitochondrial fusion/fission to cancer cell metabolism [24,25]. Apparently discordant results concerning functional impact of mitochondrial fusion/fission [2,3,4] suggest that disturbing mitochondrial dynamics is more important than specifically inhibiting fusion or fission [5].

A specific isoform of dysferlin, the closest myoferlin homolog [26], is encoded by the dysferlin_v1 alternate transcript and harbors a mitochondrial importation signal suggesting a mitochondrial localization [14]. Indeed, dysferlin was reported to interact with mitochondrial ATP synthase coupling factor 6 in HUVEC cells [27], ATP synthase subunits, and several other mitochondrial proteins in myoblasts [14]. Surprisingly, patients with dysferlinopathies caused by mutations in DYSF genes present frequently mitochondrial complex I and IV deficiencies [28]. Myoferlin was localized in several cell compartments including plasma membrane and endoplasmic reticulum [19]. However, a recent proteomic study revealed that mouse myoferlin could be present in mitochondria isolated from normal tissues [15]. For the first time, our results showed that myoferlin colocalizes and interacts with mitofusins, key proteins in mitochondrial fusion [29]. Of course, our results did not allow for excluding an indirect interaction between myoferlin and mitofusin. Interestingly, mitofusin-2 is a member of the dynamin-like GTPase superfamily and harbors a proline-rich domain [30] when myoferlin contains a SH3 domain [31] controlling protein-protein interactions through proline-rich domains. It is worth noting that a myoferlin colocalization with rab7 GTPase was reported in late endosomes [19], and a direct interaction was suggested in an exogenous expression system [32]. Myoferlin exhibits several additional domains (FerB, FerI, Ferlin C, DysF) with an unidentified function [9], opening new possibilities for its interaction with mitochondrial mitofusins, including mitofusin-1. Based on the idea that myoferlin participates functionally in the highly responsive mitochondrial dynamics through an interaction with mitofusin, this interaction should be quickly reversed, excluding probably covalent interactions. However, we do believe that this interaction has to be stable to keep mitochondria in the desired states. In this hypothesis, myoferlin could be considered as a functional interactor of mitofusins in the mitochondrial fusion (Figure 7—model A).

Endoplasmic reticulum (ER)-mitochondrial contacts were extensively studied, allowing the identification of mitochondria-associated ER membranes (MAM) working as a platform for Ca2+ transfer, lipid synthesis, and metabolism [33]. Noticeably, mitochondrial fission spatially occurs at sites of proximity to the ER [34].

Mitofusin-2 is enriched in MAM where it has been proposed to control the stability of the organelle’s interaction by homotypic and heterocomplex interaction with mitochondrial mitofusins [35]. However, an alternative model for MFN2-mediated ER-mitochondria tethering was proposed. In this model, MFN2 acts as a negative modulator of ER-mitochondria interaction, sequestering a still unknown tethering subunit [36]. Our results do not exclude an interaction of myoferlin with MFN2 on the ER. In the context of the alternative model describing the ER-mitochondria interaction, we hypothesize that myoferlin participates to the sequestration of MFN2, avoiding the stabilization of the ER-mitochondria contact needed for mitochondrial fission (Figure 7—model B).

As it stands for now, our findings clearly demonstrate a physical interaction between myoferlin and mitofusins in pancreas cancer cells. However, several points need to be clarified in the future. Among them, the identification of myoferlin and mitofusin isoforms interacting together seems to be a priority. The nature of the interaction also remains to be elucidated. Is it a direct or an indirect interaction? What are the protein domains involved? In vivo studies have to be considered, first to further validate the potential of myoferlin as a therapeutic target, and then to confirm the biological relevance of the myoferlin–mitofusin interaction. These discoveries will open up new research avenues aiming at modulating mitofusin function or targeting myoferlin to fight pancreas cancer. In this perspective, interrogating TCGA data with OncoLNC engine (http://www.oncolnc.org), it is noteworthy that mitofusin-1 (Cox coefficient = 0.353, *p* = 0.004) and myoferlin expression (Cox coefficient = 0.561, *p* < 0.001) [4,9] are both correlated with a poor overall survival in PDAC. Furthermore, Wang and coworkers reported a significant correlation between overall survival of PDAC patients and myoferlin abundance in resected tumors [37]. Myoferlin involvement in PDAC progression could go beyond cancer cell biology. Indeed, PDAC is generally considered as a “cold” immune environment probably with the participation of the Wnt/β-catenin pathway. Restoration of anti-PDAC immunity, especially in invasive tumors, remains a valid strategy [38]. In this context, we previously reported myoferlin as a negative regulator of autophagy in PDAC [4]. Additionally, a recent report pointed at myoferlin in muscle development as an indirect regulator of Wnt/β-catenin pathway by a protection of Dishevelled-2 against autophagy. The autophagy induced by myoferlin silencing could be considered as a way to promote disheveled degradation and to switch off Wnt signaling [39]. PDAC aggressiveness is due to its late diagnosis but also to its ability to acquire resistance to treatment, a process in which a Smad2/3-independent TGF-β autocrine loop [40] is involved. Interestingly, myoferlin was previously reported to regulate the TGF-β autocrine loop in breast cancer cells [41], and Smad2, a downstream transducer, was recently reported as a mitofusin interactor in mitochondrial fusion [42]. These findings raise the question of a potential interaction between mitofusin, smad2, and myoferlin and of their relevance as therapeutic target.

## 4. Materials and Methods

### 4.1. Cells and Chemicals

PaTu8988T (ACC162) were purchased from the Leibniz-Institute (DSMZ, Braunschweig, Germany) while HPNE (CRL- 4023) and C2C12 (CRL-1772) were purchased from the American Type Culture Collection (ATCC, Manassas, VA, USA). Panc-1 (CRL-1469), BxPC-3 (CRL-1687) and MiaPaCa-2 (CRL-1420) were generous gifts from, respectively, Prof. Muller and Burtea (NMR Laboratory, University of Mons, Belgium), Prof. Bikfalvi (Inserm U1029, Bordeaux, France) and Prof. De Wever (Laboratory of Experimental Cancer Research, University of Gent, Belgium). All cell lines were recently authenticated using STR profiling (DSMZ, Braunschweig, Germany). Antibodies against hemagglutinin (HA, 3724S), and GLUT1 (12939) were from Cell Signaling (Danvers, MA, USA). Vinculin (sc-25336), Myoferlin K-16 (sc-51367), MFN1 (H-65, sc-50330), SP1 (sc-17824), and HSC70 (sc-7298) antibodies were purchased from Santa-Cruz (Dallas, TX, USA). GRP78 (MAB4846) and mitochondria (Clone 113-1, MAB1273) antibodies were obtained from Millipore (Burlington, MA, USA). Myoferlin polyclonal antibody (identified here under as HPA - HPA014245) and MFN1/2 antibody (clone 3C9, ab57602) were respectively from Sigma (Bornem, Belgium) and Abcam (Cambridge, UK). All reagents were purchased from Sigma (Bornem, Belgium) unless mentioned otherwise.

### 4.2. Cell Culture

Panc-1 cells were cultured in Dulbecco’s modified Eagle’s medium (DMEM, Lonza, Basel, Switzerland) supplemented with 10% fetal bovine serum (FBS). Miapaca-2 and C2C12 were maintained in DMEM supplemented with 10% FBS, 1 mM sodium pyruvate and 4 mM L-glutamine. PaTu8988T were cultured in DMEM supplemented with 5% FBS, 5% horse serum, and 2 mM L-glutamine. BxPC-3 were maintained in RPMI1640 supplemented with 10% FBS, 1 mM sodium pyruvate, 10 mM HEPES, and 2.5 g/L of glucose. HPNE required a medium composed of 75% DMEM, 25% M3 Base (Incell Corp, San Antonio, TX, USA), 2.5% FBS, 0.01% epidermal growth factor, 2 mM L-glutamine, and 1 g/L of glucose. The cells were cultured in a humidified 5% CO2 incubator at 37 °C and were used between passage 1 and passage 10. The cells were monthly tested for mycoplasma.

### 4.3. Small Interfering RNA Transfection

Cells were transfected with 20nM siRNA using calcium phosphate. Medium was replaced 16 h after transfection, media replacement was considered as time 0. All experiments were performed 48 h after treatment. Myof#1-5′ CCCUGUCUGGAAUGAGAUUUU 3′ and Myof#2-5′ CUGAAGAGCUGUGCAUUATT 3′ siRNA were used to target myoferlin while the irrelevant siRNA-5′ CUUACGCUGAGUACUUCGAUU 3′ was used as transfection control. All siRNA were purchased from Eurogentec (Liège, Belgium).

### 4.4. Plasmid Preparation and Transfection

pCDNA3.1-Myoferlin HA [43], a plasmid encoding for human myoferlin cDNA with a C-terminal HA-tag was a gift from William Sessa (plasmid #22443; http://n2t.net/addgene:22443; RRID:Addgene_22443, Addgene, Watertown, MA, USA). The plasmid was amplified in DH5alpha bacteria cultured in classical LB medium, supplemented with ampicillin (100 µg/mL), overnight at 37 °C in an agitating incubator (200 rpm). Plasmid purification was performed using the PureYieldTM Plasmid Maxiprep System (A2393) from Promega (Fitchburg, WI, USA). Panc-1 cells were transiently transfected with 1 µg of plasmid using 2.5 µL Lipofectamine 2000 (Invitrogen, Carlsbad, CA, USA) as reported by the manufacturer’s recommendation. Medium was replaced 4 h after transfection. In order to select transfected cells, antibiotic pressure using G-418 solution was maintained at a concentration of 200 µg/mL.

### 4.5. Western Blotting

Protein samples were solubilized in 1% sodium dodecyl sulfate (SDS) supplemented with phosphatase and protease inhibitors. Bicinchoninic acid (BCA) protein assay kit (Thermo Scientific, Waltham, MA, USA) was used for protein quantification. Proteins were denatured in Laemmli’s buffer during 5 min at 99 °C. Samples were loaded on sodium dodecyl sulfate polyacrylamide gel for migration and were transferred on a PVDF membrane during 90 min at room temperature (RT) or overnight at 4 °C. Membranes were blocked for 1 h according to antibody manufacturers’ instructions. Then, they were incubated overnight at 4 °C with primary antibodies (dilution 1:1000) and probed with corresponding secondary antibodies linked to horseradish peroxidase (dilution 1:3000) for 1 h at RT. Revelation was performed using chemiluminescent reagents (ECL Western blotting substrate, Thermo Scientific, Waltham, MA, USA). Quantification was performed using densitometric analysis and immunoblots were normalized with HSC70 using ImageJ software.

### 4.6. Immunofluorescence

Cells (6 × 10^4^) were seeded on sterilized glass coverslips. After 24 h, cells were washed once with PBS and fixed with ice-cold methanol-acetone (4:1) during 10 min. Then, cells were washed twice with PBS and were blocked in 2% bovine serum albumin diluted in PBS for 30 min. After blocking, coverslips were incubated during 2 h with primary antibodies (dilution 1:100 in BSA-PBS) at RT in a humidified chamber. This step was followed by three washes in 2% BSA-PBS. Coverslips were then incubated with corresponding Alexa Fluor 488 or Alexa Fluor 546 conjugated secondary antibodies (Invitrogen, Molecular Probes, Carlsbad, CA, USA) in a humidified chamber for 45 min (dilution 1:1000 in BSA-PBS). Nuclei counterstaining was performed using hoechst DNA probe (0.01 g/L, Calbiochem, San Diego, CA, USA). Observation fields were selected randomly and pictures were acquired using an A1R confocal microscope (Nikon, Yokohama, Japan).

### 4.7. Colocalization Studies

Immunofluorescence images were deconvoluted using online NIS-elements tool (Nikon). Colocalization analysis were first performed, without selection of regions of interest, using correlation methods: Pearson coefficient correlation (PCC), Spearman’s rank correlation coefficient (SRCC), Manders’ colocalization coefficients (M1 and M2), and intensity correlation quotient (ICQ), thanks to EzColocalization [44] ImageJ plugin. Background was automatically identified according to the Costes method. Then, we used “distance between objects”-based methods: distance analysis (centers of mass of channel 1 objects inside channel 2 masks), parametric analysis of the Ripley’s K function, and non-parametric Ripley’s analysis (SODA), thanks to colocalization studio plugin in Icy software [45]. A 5-pixel maximal limit was used as a threshold.

### 4.8. Proximity Ligation Assay

Duolink PLA kit (Sigma, Bornem, Belgium) was used according to the manufacturer’s instructions. In order to detect the proximity between myoferlin and mitofusins, rabbit anti-myoferlin (HPA) and mouse anti-MFN1/2 (3C9) primary antibodies were used (dilution 1:75). Oligonucleotides conjugated secondary antibodies were provided by the kit allowing detection of a red signal if less than 40 nm separates both proteins of interest. Observation fields were selected randomly and pictures were acquired using an A1R confocal microscope. In each microscopic field, proximity dots were counted using ImageJ and divided by the number of nuclei.

### 4.9. Fluorescence Resonance Energy Transfer

All samples were proceeded as described in Immunofluorescence section. Alexa Fluor 488-conjugated secondary antibody was selected as the donor fluorophore while Alexa Fluor 546-conjugated secondary antibody was selected as the acceptor (Invitrogen, Molecular Probes, Carlsbad, CA, USA). As a positive control, two secondary antibodies were used, both targeting rabbit anti-myoferlin (HPA) primary antibody, and carrying acceptor or donor fluorophore. Finally, as a negative biological control, proteins from distinct compartments, the nuclear factor SP1 and the plasma membrane transporter GLUT1 were selected. Images were acquired with a LSM880 Airyscan Elyra Microscope (Zeiss, Oberkochen, Germany).

### 4.10. Co-Immunoprecipitation

Proteins were extracted using non-denaturing buffer containing Tris-HCl pH 8 (20 mM), NaCl (137 mM), NP40 (1%), EDTA (2 mM), and supplemented with protease inhibitors. Following extraction, proteins were incubated under rotation at 4 °C during 30 min and were centrifuged at 14000× g for 15 min at 4 °C to eliminate cell debris. 5 µg of MFN1/2 or isotype IgG control (Thermo Scientific, Waltham, MA, USA) antibodies were incubated overnight with 500 µg of the protein extract. Then, protein A/G magnetic beads (Thermo Scientific, Waltham, MA, USA) were added and incubated at 4 °C under rotation for 2 h. After three washes with a low salt buffer containing SDS (0.1%), Triton X-100 (1%), EDTA (2 mM), Tris-HCl pH 8 (20 mM) and NaCl (150 mM), and one wash of high salt buffer composed of SDS (0.1%), Triton X-100 (1%), EDTA (2 mM), Tris-HCl pH 8 (20 mM), and NaCl (450 mM), proteins were eluted from magnetic beads using Laemmli’s buffer and then processed for Western blotting.

### 4.11. Mitochondrial Enrichment

Mitochondrial isolation kit (Qiagen, Hilden, Germany) was used according to the manufacturer’s instructions. Briefly, 2 × 10^7^ washed cells were suspended in lysis buffer in order to disrupt plasma membrane. After centrifugation (1000× *g*, 10 min, 4 °C), the supernatant contained cytosolic proteins while the pellet was composed of intact mitochondria, endoplasmic reticulum, and other compartmentalized organelles. The pellet was suspended in disruption buffer and homogenized in a potter with a glass pestle (15X). After centrifugation (1000× *g*, 10 min, 4 °C), nuclei, cell debris, and unbroken cells were pelleted while the microsomes and mitochondria were contained in the supernatant. The supernatant was then centrifuged (6000× g, 10 min, 4 °C) and the obtained pellet was resuspended in mitochondrial purification buffer. The mitochondrial suspension was pipetted on density gradient layers. After centrifugation (14,000× *g*, 10 min, 4 °C), mitochondria were pelleted and harvested. All fractions were processed for further Western blotting experiments.

### 4.12. TMRE Mitochondrial Staining

Tetramethylrhodamine ethyl ester (TMRE) was used to visualize mitochondria in living Panc-1 cells. siRNA-transfected cells were seeded in 8-well slides (IBIDI, Munich, Germany) at low confluence. Staining was performed for 15 min at 37 °C with TMRE (1 nM). Images were acquired by epifluorescence microscopy as Z-stacks with an A1R microscope.

### 4.13. Ultrastructural Analysis

Panc-1 cells were fixed for 90 min at room temperature with glutaraldehyde (2.5%) in a Sörensen phosphate buffer (0.1 M, pH 7.4) and post-fixed for 30 min with 2% osmium tetroxide. Embedding and observation were performed as previously described [12].

### 4.14. Oxygen Consumption Rate Analysis

Oxygen consumption rates were measured with a Seahorse XFp extracellular flux analyzer (Agilent, Santa Clara, CA, USA). siRNA-transfected Panc-1 cells were seeded (13000 cells per well) in XFp mini-plates and allowed to attach overnight. For mitochondrial OCR analysis, cells were kept in unbuffered serum-free DMEM (Basal DMEM, Agilent) supplemented with pyruvate (1 mM), glutamine (2 mM), glucose (10 mM), pH 7.4 at 37 °C, and ambient CO2 for 1 h before the assay. During the assay, cells were successively stressed with oligomycin (1 µM), FCCP (1.0 µM), and rotenone/antimycin A (0.5 µM each) mix. Results were normalized according to the cell number evaluated by Hoechst (2 µg/mL) incorporation after cold methanol/acetone fixation.

### 4.15. Statistical Analysis

Results were presented as individual scatter-plots together with median and interquartile range. Two-sided statistical analysis was performed using non-parametric analysis of variance. Groups were compared by Dunn’s *t* test, and *p* < 0.05 was considered as statically significant.

## 5. Conclusions

The disruption of the mitochondrial network is associated with a significant inhibition of cell proliferation and migration in pancreas cancer cells. The discovery of a mitofusin–myoferlin interaction in PDAC cell lines opens up new research avenues aiming at modulating mitofusin function in pancreas cancer.

## Figures and Tables

**Figure 1 cancers-12-01643-f001:**
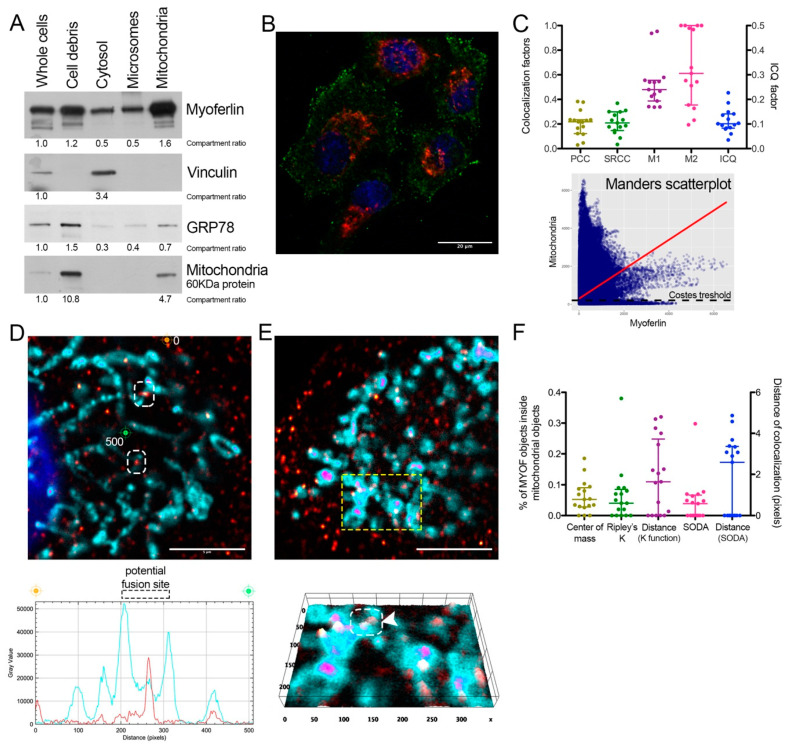
Myoferlin was colocalized with mitochondria in Panc-1 cells. (**A**) Western blot of 6 µg protein samples from whole Panc-1 cells and several cellular compartments isolated from Panc-1 cells. Myoferlin, vinculin, GRP78, and a 60 kDa mitochondrial protein were detected on the same membrane. Compartment relative quantification was performed using ImageJ; (**B**) representative confocal image of nuclei (blue), myoferlin (K-16—green) and mitochondria (113-1—red) immunofluorescence. Scale bar = 20 µm; (**C**) Pearson (PCC), Spearman rank (SRCC) correlation coefficients, Manders’ colocalization coefficients (M1,M2), and intensity correlation quotient (ICQ) calculated on 17 independent microscopic fields. Manders scatterplot, associated with its linear regression (red line), shows the correlation between the intensity of each pixels in each channel. (**D**,**E**) Deconvoluted confocal image of nuclei (blue), myoferlin (K-16—“hot” red scale), mitochondria (113-1—“cold” cyan scale). Scale bar = 5 µm. Regions surrounded by white dashed boxes are putative mitochondrial fusion sites. (**D**) Channel intensity profile was established following the segment between orange (0-pixel position) and green (500-pixel position) cross marks; (**E**) The region surrounded by a yellow dashed box was used to generate the 2D intensity profile. Regions surrounded by white dashed box and marked by white arrow head is a putative mitochondrial fusion site; (**F**) percentage of myoferlin-positive objects (*N* = 4286) with the center of a mass overlapping mitochondrial object (*N* = 459), a percentage of myoferlin-positive object colocalizing mitochondrial object calculated by fitting of the Ripley’s K function or by statistical object distance analysis (SODA). Colocalization distances in pixels were measured in both cases. All experiments were performed as three independent biological replicates.

**Figure 2 cancers-12-01643-f002:**
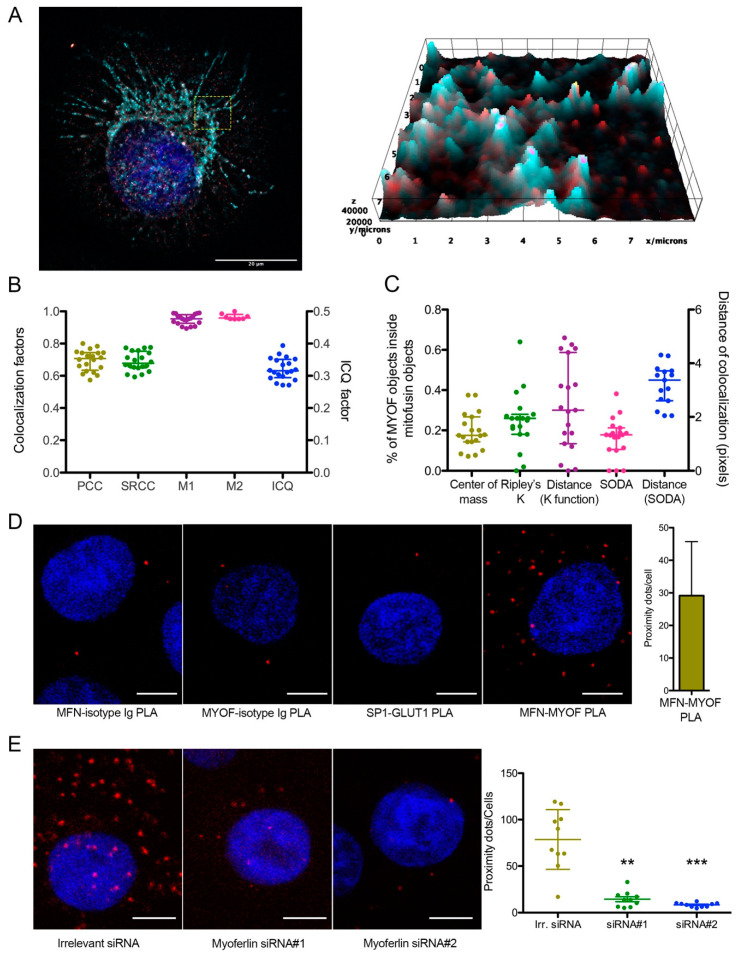
Myoferlin was colocalized with mitochondrial fusion machinery. (**A**) Representative deconvoluted confocal image of nuclei (blue), myoferlin (K16—“hot” red scale) and mitofusin-1 (H65—“cold” cyan scale) immunofluorescence. Scale bar = 20 µm. Region surrounded by yellow dashed box was used to generate the 2D intensity profile; (**B**) Pearson (PCC), Spearman rank (SRCC) correlation coefficients, Manders’ colocalization coefficients (M1,M2), and intensity correlation quotient (ICQ) were calculated on 20 independent microscopic fields randomly selected; (**C**) percentage of myoferlin-positive objects (*N* = 7128) with center of mass overlapping mitochondrial object (*N* = 369), percentage of myoferlin-positive object colocalizing mitochondrial object calculated by fitting of the Ripley’s K function or by statistical object distance analysis (SODA). Colocalization distances in pixels were measured in both cases; (**D**) representative images of proximity ligation assay (PLA) between myoferlin (HPA) and mitofusin-1/2 (3C9). Scale bar = 4 µm. Controls were established by substitution of antibodies by control isotypes or by using antibodies against non-interacting proteins (SP1 and GLUT1); (**E**) representative images of PLA in Panc-1 cells transfected with irrelevant or myoferlin-specific siRNA. Scale bar = 4 µm. MFN1/2-MYOF PLA (*N* = 10) were quantified using ImageJ. Kruskal–Wallis non-parametric test followed by Dunn’s pairwise comparison was performed, ** *p* < 0.01, *** *p* < 0.001. All experiments were performed as three independent biological replicates.

**Figure 3 cancers-12-01643-f003:**
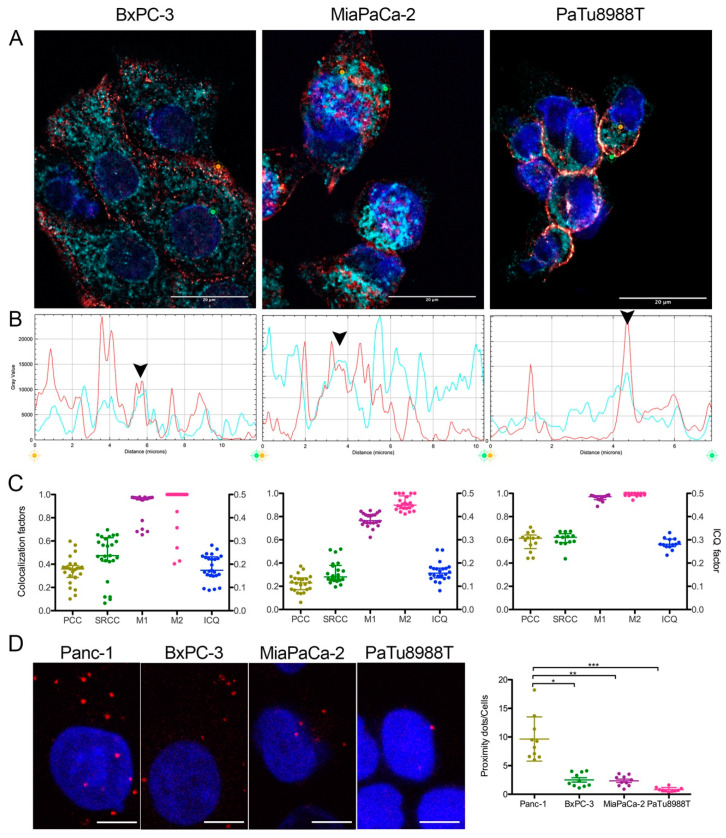
Myoferlin was colocalized with MFN1/2 in several pancreas cancer cell lines. (**A**) representative deconvoluted confocal image of nuclei (blue), myoferlin (HPA—“hot” red scale) and mitofusin-1/2 (3C9—“cold” cyan scale) immunofluorescence of BxPC-3, MiaPaCa-2 and PaTu8988T cell lines. Scale bar = 20 µm. (**B**) Channel intensity profiles were established following the segment between orange and green cross marks. Black arrow heads indicate colocalization spots. (**C**) Pearson (PCC), Spearman rank (SRCC) correlation coefficients, Manders’ colocalization coefficients (M1,M2), and intensity correlation quotient (ICQ) were calculated on >13 independent microscopic fields. (**D**) Representatives images of MFN1/2-MYOF proximity ligation assay (PLA). Scale bar = 4 µm. MFN1/2-MYOF PLA (*N* = 10) were quantified using ImageJ. Kruskal–Wallis non-parametric test followed by Dunn’s a pairwise comparison was performed, * *p* < 0.05, ** *p* < 0.01, *** *p* < 0.001. All experiments were performed as three independent biological replicates.

**Figure 4 cancers-12-01643-f004:**
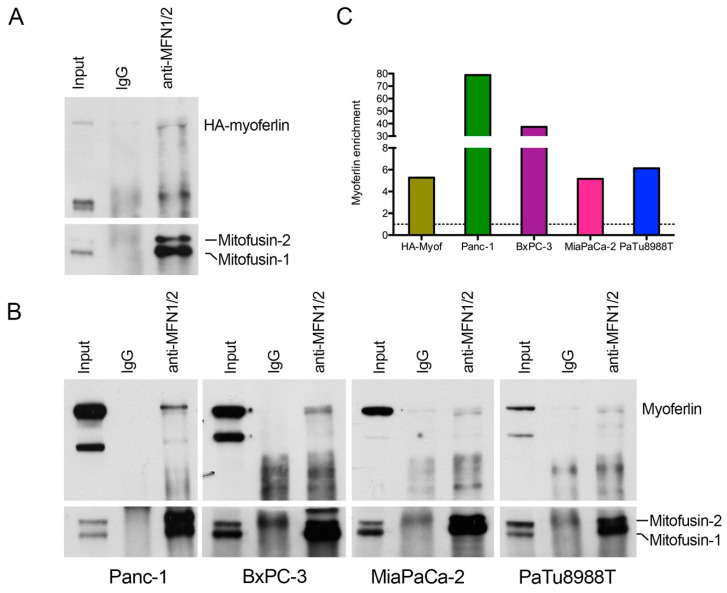
Myoferlin interacts with mitofusins in pancreas cancer cell lines. (**A**) coimmunoprecipitation of mitofusins and HA-tagged myoferlin with an anti-mitofusin antibody. Western blot of protein samples from whole cells (input), IgG control immunoprecipitation (IgG), and mitofusins immunoprecipitation (anti-MFN1/2) of HA-myoferlin transfected Panc-1 cells. HA-myoferlin and mitofusins were detected on the same membrane; (**B**) coimmunoprecipitation of mitofusins and endogenous myoferlin with an anti-mitofusin antibody. Western blot of protein samples from whole cells, IgG control immunoprecipitation, and mitofusins immunoprecipitation of Panc-1, BxPC-3, MiaPaCa-2, and PaTu8988T cell lines. Myoferlin and mitofusins were detected on the same membrane; (**C**) myoferlin (or HA-tagged myoferlin) enrichment in anti-MFN1/2 relatively to IgG. Quantification was performed using ImageJ. All experiments were performed as three independent biological replicates.

**Figure 5 cancers-12-01643-f005:**
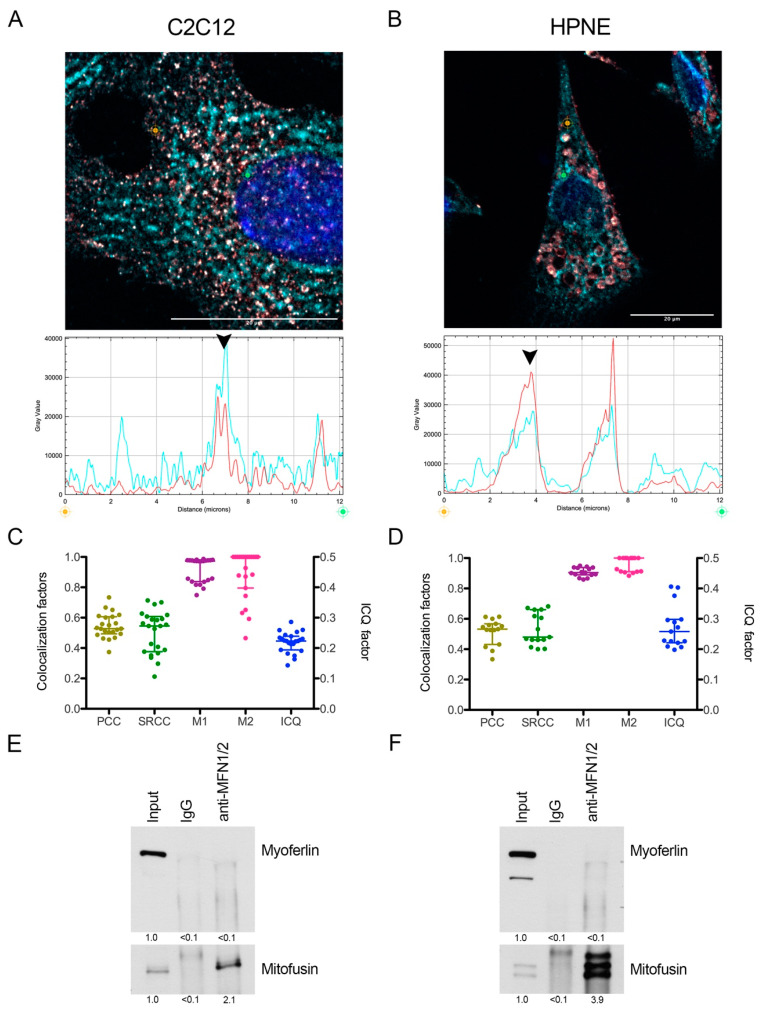
Myoferlin was colocalized with mitofusin-1 in normal cell lines. Representative deconvoluted confocal image of nuclei (blue), myoferlin (HPA—“hot” red scale) and mitofusin-1/2 (3C9—“cold” cyan scale) immunofluorescence of (**A**) C2C12 murine myoblast and (**B**) immortalized human pancreatic normal epithelial (HPNE) cell lines. Scale bar = 20 µm. Channel intensity profiles were established following the segment between orange and green cross marks. Black arrow heads indicate colocalization spots. (**C**,**D**) Pearson (PCC), Spearman rank (SRCC) correlation coefficients, Manders’ colocalization coefficients (M1,M2), and intensity correlation quotient (ICQ) were calculated on >15 independent microscopic fields; (**E**,**F**) coimmunoprecipitation of mitofusins and endogenous myoferlin with an anti-mitofusin antibody. Western blot of protein samples from whole cells (input), IgG control immunoprecipitation (IgG), and mitofusins immunoprecipitation (anti-MFN1/2) of C2C12 and HPNE cell lines. Myoferlin and mitofusins were detected on the same membrane. Quantification was performed using ImageJ. All experiments were performed as three independent biological replicates.

**Figure 6 cancers-12-01643-f006:**
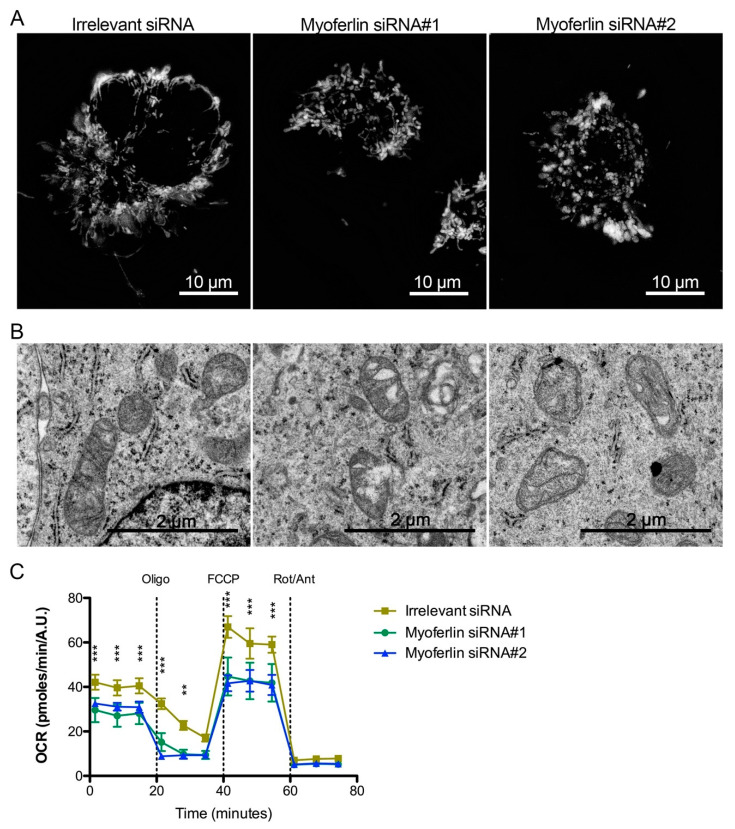
Mitochondrial impact of myoferlin deletion in pancreas cancer cells. (**A**) Mitochondria were stained with tetramethylrhodamine ethyl ester (1 nM TMRE) in Panc-1 living cells depleted for myoferlin. (**B**) Panc-1 cells depleted for myoferlin were fixed with glutaraldehyde and observed under transmission electron microscope. (**C**) Kinetic oxygen consumption rate (OCR) response of Panc-1 cells to oligomycin (oligo, 1 µM), FCCP (1.0 µM), rotenone, and antimycin A mix (Rot/Ant, 0.5 µM each). Each data point represents mean ± SD of technical replicates. All experiments were performed as three independent biological replicates. *** *p* < 0.001, ** *p* < 0.01.

**Figure 7 cancers-12-01643-f007:**
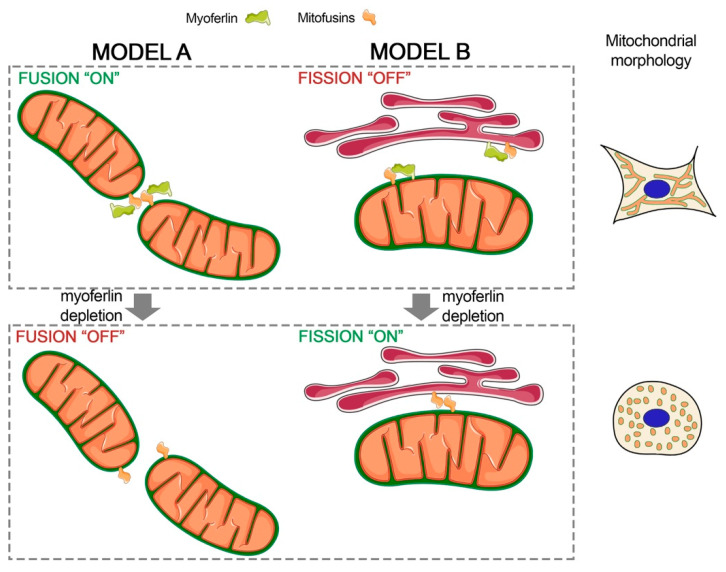
Proposed models for myoferlin involvement in mitochondrial dynamics. Model A describes the functional interaction between mitofusins and myoferlin. Myoferlin interacts with mitofusin and participates, as a positive regulator, to mitochondrial fusion. Myoferlin depletion reduces efficiency or inhibits the mitofusin-mediated mitochondrial fusion. Model B depicts the mitofusin sequestration by myoferlin impairing the ER-mitochondria tethering and subsequent fission. Myoferlin silencing results in the stabilization of the ER-mitochondria tethering by mitofusin interaction allowing ER wrapping and DRP1 recruitment. Designed with Servier Medical Art (https://smart.servier.com) licensed under a Creative Commons Attribution 3.0 Unported License.

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
