# Peer review of "Myoferlin Is a Yet Unknown Interactor of the Mitochondrial Dynamics’ Machinery in Pancreas Cancer Cells"

_cancers, 2020, doi:10.3390/cancers12061643_

Round 1

Reviewer 1 Report

This is a study about myoferlin. The authors demonstrate that myoferlin expression leads to mitochondrial fusion in cultured pancreatic cancer cells. In addition, since no interaction could be confirmed in normal cells, it is concluded that it is a characteristic in cancer cells. It is very easy to understand, and I would like to ask you to discuss the function of myoferlin and its relationship with cancer cells (especially, with regard to tumor malignancy). I also want to know the future perspective of the authors.

Author Response

Authors want to thank reviewer for his/her comments regarding the submitted manuscript. As requested, we add a paragraph in discussion section describing briefly the physiological role of myoferlin and connecting it to cancer cells. We also cited two recently published reviews on the topic (Bulankina et al., 2020 and Dong et al., 2019). A perspective section was also added at the end of the manuscript. These additions improve greatly our manuscript.

Reviewer 2 Report

The authors elegantly demonstrated for the first time that myoferlin colocalizes with mitofusins in a PDAC cell line. However, we are not thoroughly convinced about their interactions.  Common techniques like IP/Co-IP/Pull-down assay cannot discriminate between the direct and indirect protein-protein interaction. Therefore, we would like to make few comments below highlighting the subject.

  1. We are wondering whether performing a Fluorescence Resonance Energy Transfer (FRET) analysis would be more informative regarding the interaction of the myoferlin and mitochondria-specific proteins. This is because the co-precipitation itself does not necessarily guarantee that there would be a meaningful protein-protein (myoferlin-mitofusion) interaction. However, for FRET to be functional, the two interacting proteins need to be in less than 10nm proximity.
  2. The authors showed that there is a maximal 40 nm distance between myoferlin and MFN1/2 (Figure 3D) using the proximity ligation assay. The finding also raise a question whether the two proteins are close enough to interact physically or they are part of a signaling cascade where interactions to be mediated by an intermediate linking/linker molecule.
  3. We appreciate the authors’ effort to propose the models for myoferlin involvement in mitochondrial dynamics depicted in Figure 7. However, they could already provide preliminary results supporting the model by examining the mitochondrial morphology for fusion/fission using electron microscopy. In addition, we ask the authors for additional experiments using siRNA-mediated knockdown of myoferlin to study a) changes in cellular respiration by measuring OXPHOS and b) consequent impact on mitophagy that results from switching on/off the fusion/fission. We request these crucial experiments to the authors as they prove the functional interactions the authors hypothesized. Without these fictional readouts, the proof of co-localization of myoferlin and mitofusion carry minimal biological significance.
  4. For the robustness of the reproducibility, we would like to know how many times the in vitro study was repeated. A minimum acceptable number is 3. If the criterion is not met, we request for additional round of experiments.

Minor comments:

  1. The order of the Pictures in Figure 1 C to F is quite confusing, We would recommend another Layout, maybe A, B, C in the first row, than D and E and in the last row maybe F, or D, E and F in one row.
  2. For figure 1A, the authors mentioned in figure legend about performing a relative quantification of the protein bands using ImageJ. However, the quantitative data was not shown in 1A. We request the authors to provide this additional data.
  3. In Figure 1, the authors demonstrated that endogenous myoferlin is present in mitochondrial crude extract and co-localized partly with mitochondria. This was conducted using the ‘distance between objects-based method’. The same experiment was repeated in Figure 2 with a different antibody. This validation should go to the supplement as this a validation of Figure 1.
  4. In Figure 3B, we would suggest to clarify how those 20 independent fields were chosen. Is it randomly chosen or by following certain criteria-this needs to be mentioned.
  5. High-resolution images are required for figure 3 (D, E). The knockdown of myoferlin is not clearly visible. Better images also required for Figure 4D.
  6. Figure 5 (C) is discussed in the text and in the figure legend but the figure C is absent from Figure 5. We request the author to add it.
  7. For a future study, we would suggest experiments on PDAC mouse models, to proof that the results also apply in vivo and not limited in an in vitro set up.
  8. About the poposed myoferlin-mitofusion interactions, we are curious to know what could be the type of type of this protein-protein interaction. Could it be a stable interaction vs. transient interaction? Furthermore, is it covalent vs. non-covalent? We would like to hear the authors thought on it as this might shed light on how the mitochondrial dynamics being regulated by their interaction.

Author Response

REVIEWER 2

The authors elegantly demonstrated for the first time that myoferlin colocalizes with mitofusins in a PDAC cell line. However, we are not thoroughly convinced about their interactions.  Common techniques like IP/Co-IP/Pull-down assay cannot discriminate between the direct and indirect protein-protein interaction. Therefore, we would like to make few comments below highlighting the subject.

  1. We are wondering whether performing a Fluorescence Resonance Energy Transfer (FRET) analysis would be more informative regarding the interaction of the myoferlin and mitochondria-specific proteins. This is because the co-precipitation itself does not necessarily guarantee that there would be a meaningful protein-protein (myoferlin-mitofusion) interaction. However, for FRET to be functional, the two interacting proteins need to be in less than 10nm proximity.

Authors thank the reviewer for his/her sound comments. Indeed, our results do not discriminate indirect from direct protein-protein interaction. We can only assume a maximal 40 nm proximity between myoferlin/mitofusins, and an undefined interaction. As suggested by the reviewer, FRET analysis would probably be more informative if performed with YFP-myoferlin & CFP-mitofusin fusion proteins. However, the 2061 amino acid length of myoferlin is per se a limitation. Indeed, a theoretical protein size calculation model (http://www.calctool.org/CALC/prof/bio/protein_size), based on a globular protein shape, proposed an 8.7 nm length for myoferlin. Owing to the non-globular shape of myoferlin, we can reasonably assume a size larger than 10 nm. Depending of the myoferlin’s domain interacting with mitofusin, we cannot exclude that FRET give false negative results. Adding flexibility to the FRET settings by using antibodies conjugated with donor/acceptor fluorophores would probably alleviate this difficulty but will increase accordingly the maximal detection distance between interacting proteins. Therefore, we would obtain a detection limit similar to that of PLA which would lose interest.

The period of 10 days to carry on this revision does not allow the implementation of a direct-FRET experiment from scratch (construction of myoferlin-YFP and mitofusin-CFP fusion proteins, the selection of the best isoforms, transfection and selection of cells, …). We thus decided to perform an indirect-FRET using secondary antibodies conjugated with Alexafluor 488 and Alexafluor 546. SP1/GLUT1 staining was used as a negative control. One primary antibody for myoferlin (HPA) and two different secondary antibodies fused with donor and acceptor fluorophores respectively were used in positive control (Guala et al., 2018 - doi: 10.1088/2050-6120/aab932). Results showed a FRET signal in positive control and in myoferlin/mitofusin double staining. Unfortunately, it is not possible to confirm a direct interaction since we performed indirect FRET. However, these results confirmed the proximity ligation assay and were include as supplemental information (figure S3).

  1. The authors showed that there is a maximal 40 nm distance between myoferlin and MFN1/2 (Figure 3D) using the proximity ligation assay. The finding also raise a question whether the two proteins are close enough to interact physically or they are part of a signaling cascade where interactions to be mediated by an intermediate linking/linker molecule.

Authors agree with this very pertinent remark. As stated in answer to comment #1, based on the maximal 40 nm proximity between myoferlin/mitofusins, and on the immunoprecipitation assays, we cannot exclude indirect interaction between myoferlin and mitofusin. Consequently, we decided to temper our conclusion by mentioning this eventuality.

  1. We appreciate the authors’ effort to propose the models for myoferlin involvement in mitochondrial dynamics depicted in Figure 7. However, they could already provide preliminary results supporting the model by examining the mitochondrial morphology for fusion/fission using electron microscopy. In addition, we ask the authors for additional experiments using siRNA-mediated knockdown of myoferlin to study a) changes in cellular respiration by measuring OXPHOS and b) consequent impact on mitophagy that results from switching on/off the fusion/fission. We request these crucial experiments to the authors as they prove the functional interactions the authors hypothesized. Without these fictional readouts, the proof of co-localization of myoferlin and mitofusion carry minimal biological significance.

Authors thank the reviewer for his/her suggestion regarding the addition of results supporting the proposed model. We added observation of mitochondrial network in live cell (TMRE staining followed by epifluorescence microscopy), ultrastructural analysis of mitochondria by electron microscopy, and oxygen consumption rate analysis using both myoferlin siRNA (Figure 6). These results showed clearly a swelling of mitochondria, a disruption of the mitochondrial network, an alteration of matrix and cristae morphology and a reduction of oxygen consumption rate, supporting our model and the biological relevance of myoferlin in mitochondrial function.

  1. For the robustness of the reproducibility, we would like to know how many times the in vitro study was repeated. A minimum acceptable number is 3. If the criterion is not met, we request for additional round of experiments.

Authors apologize for the lack of information in the original manuscript and thank reviewer for his/her careful observation. All experiments were performed as three independent biological replicates. This information was added in figure legends.

Minor comments:

  1. The order of the Pictures in Figure 1 C to F is quite confusing, We would recommend another Layout, maybe A, B, C in the first row, than D and E and in the last row maybe F, or D, E and F in one row.

Authors thanks the reviewer for the pertinent suggestion. Figure 1 panels were reordered as ABC in row 1 and DEF in row 2.

  1. For figure 1A, the authors mentioned in figure legend about performing a relative quantification of the protein bands using ImageJ. However, the quantitative data was not shown in 1A. We request the authors to provide this additional data.

Authors thanks reviewer for this comment. Compartment quantification relative to whole cells sample is shown under each western-blot. Relative quantification was also added to the supplemental Figure S4.       

  1. In Figure 1, the authors demonstrated that endogenous myoferlin is present in mitochondrial crude extract and co-localized partly with mitochondria. This was conducted using the ‘distance between objects-based method’. The same experiment was repeated in Figure 2 with a different antibody. This validation should go to the supplement as this a validation of Figure 1.

Authors thanks the reviewer for the relevant suggestion. Original Figure 2 was moved to supplement (supplemental figure S1).

  1. In Figure 3B, we would suggest to clarify how those 20 independent fields were chosen. Is it randomly chosen or by following certain criteria-this needs to be mentioned.

Authors apologize for the lack of information in the materials and methods section and thank reviewer for his/her careful observation. All microscopy fields were randomly selected. This point was added in the immunofluorescence section.

  1. High-resolution images are required for figure 3 (D, E). The knockdown of myoferlin is not clearly visible. Better images also required for Figure 4D.

Authors thanks reviewer for his/her comment. PLA panels were reconstructed starting from high resolution images. We do believe that PDF conversion altered image quality. Additionally, we proposed to include full size figures as supplement.

  1. Figure 5 (C) is discussed in the text and in the figure legend but the figure C is absent from Figure 5. We request the author to add it.

Once more, authors thank reviewer for his/her careful proofreading. The figure 5C citation was a typo error and should have been supplemental figure S4. The correction was implemented in the reviewed version of the manuscript.

  1. For a future study, we would suggest experiments on PDAC mouse models, to proof that the results also apply in vivo and not limited in an in vitro set up.

Authors thanks the reviewer for the relevant suggestion. Indeed, in vivo experiments are the ultimate preclinical proofs. This point was added in a short perspective section at the end of manuscript.

  1. About the poposed myoferlin-mitofusion interactions, we are curious to know what could be the type of type of this protein-protein interaction. Could it be a stable interaction vs. transient interaction? Furthermore, is it covalent vs. non-covalent? We would like to hear the authors thought on it as this might shed light on how the mitochondrial dynamics being regulated by their interaction.

Authors thanks reviewer for this invitation to expose our thought about the nature of myoferlin-mitofusin interaction. Based on the idea that myoferlin participates functionally to the highly responsive mitochondrial dynamics through an interaction with mitofusin, this interaction should be quickly reversed, excluding probably covalent interactions. Accordingly, we mentioned in the manuscript a potential SH3-PR domains interaction. We guess this interaction has to be stable to keep mitochondria in the desired states. This point was included in the discussion.

Reviewer 3 Report

Sandi Anania and colleagues report about the role of myoferlin, mitochondria and mitofusin; pancreatic cancer as potential targets to envision novel theragnostic strategies. The data are of interest. Nonetheless, a few points need to be discussed in order to reach the level of pieces of evidence required from a journal like Cancers.

Figure1, 5 and 6. for all Western blot figures, densitometry readings/intensity ratio of each band should be included; the whole Western blot showing all bands and molecular weight markers should be included in the Supplementary Materials;

Gene silencing: gene silencing experiments should use at least two gene-specific siRNAs. The authors correctly accomplished such a standard. Nonetheless, do the authors considered an shRNA or CRISPR/CAS based strategies? This might be beyond the scope of this manuscript, nevertheless, the authors comment on this topic would be worth.

General comment: do the author had the chance to verify the clinical impact of their findings? Indeed, while interrogating in silico the TCGA study public data, the 5-year survival of myoferlinhigh patients statistically significantly differs from OS of myoferlinlow subjects (P score 0.00014). Remarkably, in silico analysis also provide evidence that mitofusin impacts pts survival in much the same way (P score 0.0038).

Did the author confirm those findings? Alternatively, while assuming these as potential study limitations, a comment from the author can be worth in order to highlight the value and the limitations of the data presented, interpreting them as hypothesis-generating data that need to be confirmed in a prospective fashion.

In the frame of this clinical translational impact, some of the above-mentioned annotations can significantly improve the manuscript quality and provide important information for the scientific community. MIAPaCa-2 cells have been uncovered by the authors to be relevantly overexpressing mitofusin and myoferlin. I think it is important to mention that myoferlin can protect against autophagy-mediated degradation, thus promoting activation of the Wnt/β-catenin signaling pathway. Because of these intimate interactions, the capacity of dendritic cells and endothelial cells ECs as antigen-presenting cells (APC) can be also discussed, since several examples have been recently published (i.e. PMID: 31277479), especially while explaining MIAPaCa-2 cells neoplastic phenotype.

Moreover, as an interesting candidate for targeting the integral mitochondrial protein required for mitochondrial fusion mitofusin can interact with the cytoplasmic-localized SMAD2, while promoting mitochondrial fusion. This is in stark contrast to the previously established role of TGF-β-dependent mitochondrial fission, a process largely mediated by pro-apoptotic transcriptional responses to TGF-β-induced SMAD2/3 signaling. These findings highlight the fundamental potential role of mitofusin in the vicious cycle potentially characterizing the crosstalk with the microenvironment (PMID: 30866547). It would be worth to receive a comment from the authors regarding this topic, uncovered to point towards a potential Achilles’ heel of PDAC that might be exploited therapeutically in the future.

Author Response

Sandy Anania and colleagues report about the role of myoferlin, mitochondria and mitofusin; pancreatic cancer as potential targets to envision novel theragnostic strategies. The data are of interest. Nonetheless, a few points need to be discussed in order to reach the level of pieces of evidence required from a journal like Cancers.

Figure1, 5 and 6. for all Western blot figures, densitometry readings/intensity ratio of each band should be included; the whole Western blot showing all bands and molecular weight markers should be included in the Supplementary Materials;

Authors thanks reviewer for this comment. Compartment quantification relative to whole cells sample is shown under each western-blot in figure 1. Relative quantification (HSC70 was used for normalization) were added to the supplemental Figure S4. Myoferlin (or HA-tagged myoferlin) enrichment in anti-MFN1/2 immunoprecipitation relatively to IgG was represented as a bar graph in figure 4C (previously figure 5) and as densitometry values in figure 5EF (previously figure 6). Additionally, full western blots were included in supplement.

Gene silencing: gene silencing experiments should use at least two gene-specific siRNAs. The authors correctly accomplished such a standard. Nonetheless, do the authors considered an shRNA or CRISPR/CAS based strategies? This might be beyond the scope of this manuscript, nevertheless, the authors comment on this topic would be worth.

Authors thanks the reviewer for the relevant suggestion. Indeed, we have considered both strategies. Panc-1 expressing myoferlin shRNAs are available in our Lab. Our analysis revealed that shortly after shRNA expression with lentivirus, myoferlin silencing was effective with the same effects than siRNAs on oxygen consumption rate and on cell growth decrease. Concerning CRISPR/CAS9 things were different. We were unable to obtain any myoferlin KO clones. This result suggests the necessity to keep a minimal myoferlin abundance to allow cell growth.

General comment: do the author had the chance to verify the clinical impact of their findings? Indeed, while interrogating in silico the TCGA study public data, the 5-year survival of myoferlinhigh patients statistically significantly differs from OS of myoferlinlow subjects (P score 0.00014). Remarkably, in silico analysis also provide evidence that mitofusin impacts survival in much the same way (P score 0.0038).

Did the author confirm those findings? Alternatively, while assuming these as potential study limitations, a comment from the author can be worth in order to highlight the value and the limitations of the data presented, interpreting them as hypothesis-generating data that need to be confirmed in a prospective fashion.

Reviewer raised a very important point. Indeed, TCGA-PAAD data showed a significant difference in OS between myoferlinlow/high subjects and between mitofusin1low/high subjects. We previously published the correlation between Myoferlin expression and OS (Rademaker et al., 2018 - doi: 10.1038/s41388-018-0287-z& Peulen et al., 2019 - doi: 10.3390/cells8090954). Unfortunately, the small size of the cohort (n=40) in our retrospective study did not allow to reach a significant association between myoferlin abundance and OS. Wang and coworkers published (doi: 10.1016/j.jprot.2013.06.032) a retrospective study with 154 PDAC patients. They reported a significant OS difference between myoferlin-high and myoferlin-low subjects. A part of the discussion has been dedicated to this specific aspect.

In the frame of this clinical translational impact, some of the above-mentioned annotations can significantly improve the manuscript quality and provide important information for the scientific community. MIAPaCa-2 cells have been uncovered by the authors to be relevantly overexpressing mitofusin and myoferlin. I think it is important to mention that myoferlin can protect against autophagy-mediated degradation, thus promoting activation of the Wnt/β-catenin signaling pathway. Because of these intimate interactions, the capacity of dendritic cells and endothelial cells ECs as antigen-presenting cells (APC) can be also discussed, since several examples have been recently published (i.e. PMID: 31277479), especially while explaining MIAPaCa-2 cells neoplastic phenotype.

Authors thank reviewer for his/her sound remark and very original idea. Indeed, MiaPaCa-2 cell line seems to express myoferlin as highly as Panc-1 and BxPC-3 cell lines did (supplemental figure S4). Myoferlin silencing was previously reported by our team as a positive regulator of autophagy in PDAC cells (Rademaker et al., 2018 - doi: 10.1038/s41388-018-0287-z). Additionally, in muscle, myoferlin was reported to stabilize Dishevelled-2 against autophagy, allowing Wnt signaling (Han et al., 2019 - doi: 10.3390/ijms20205130). As such, we include a discussion paragraph proposing to target myoferlin to restore anti-PDAC immunity by an autophagy-mediated regulation of Wnt pathway.

Moreover, as an interesting candidate for targeting the integral mitochondrial protein required for mitochondrial fusion mitofusin can interact with the cytoplasmic-localized SMAD2, while promoting mitochondrial fusion. This is in stark contrast to the previously established role of TGF-β-dependent mitochondrial fission, a process largely mediated by pro-apoptotic transcriptional responses to TGF-β-induced SMAD2/3 signaling. These findings highlight the fundamental potential role of mitofusin in the vicious cycle potentially characterizing the crosstalk with the microenvironment (PMID: 30866547). It would be worth to receive a comment from the authors regarding this topic, uncovered to point towards a potential Achilles’ heel of PDAC that might be exploited therapeutically in the future.

Authors thank reviewer for his/her suggestion. Although very interesting, this topic appears far beyond the context of our manuscript. Anyway, authors have included few lines on a possible link between mitofusin, myoferlin and smad2 in the discussion section.

Round 2

Reviewer 2 Report

Major comment:

We would like to thank the authors for sincerely addressing our concerns and suggestion reflected in their effort to perform the indirect FRET to confirm Myoferlin-Mitofusin interaction. We also appreciate the additional experiments we requested to strengthen/validate theirs findings on Myoferlin as an important regulator of mitochondrial dynamics. Given the limited amount of time to revise the entire manuscript, the authors did their best with high efficiency.

Minor comment:

For figure 3D, E We would suggest to take larger field (that only shows few cells) at higher magnification for better visualization of the image.

Author Response

Authors thanks reviewer for his/her benevolent comments which greatly improve the quality of our manuscript. Indeed, we did our best to satisfy your justified requests, and first authors did a tremendous job, especially in the current health context.

Concerning this second revision round, we have adapted all PLA figures (Figure 2D-E, Figure 3D) as requested. Figure legends were adapted accordingly.

Reviewer 3 Report

The authors addressed all my requests. I have no further comments.

Author Response

Authors thank reviewer for his/her carefully revision our their manuscript.